# Evaluation of an evidence-based practice continuing education course for Canadian Naturopathic Doctors

Monique Aucoin[ID][1,2]*, Genevieve Newton[2], Matthew Leach[ID][3], Kieran Cooley[ID][1,3,4,5]

1 Canadian College of Naturopathic Medicine, 2 University of Guelph, 3 Southern Cross University, 4 University of Technology Sydney, 5 University of Toronto

* maucoin@ccnm.edu

## Abstract

### Background

Evidence-based practice (EBP) combines the best available evidence with clinician expertise and patient preference to improve patient outcomes. Recent evidence indicates that Canadian Naturopathic Doctors (NDs) are interested in EBP skill development. The primary objective of the present study was to assess the feasibility and acceptability of a co-designed EBP Continuing Education (CE) course for Canadian NDs. Secondary objectives included assessing changes in EBP skill, use, attitudes, and knowledge.

### Methods

The CE course was evaluated using a pre-post design involving licensed Canadian NDs. The CE course consisted of five weekly, one-hour sessions that were delivered virtually. On course completion, participants reported on their level of satisfaction and provided suggestions for improvement. EBP skill, attitudes and use were assessed using the validated Evidence-Based Practice Attitudes and Utilization Survey. EBP knowledge was objectively assessed using a quiz. Changes in EBP skill, attitudes, use and knowledge were compared between baseline and the end of the course. Use of evidence was reassessed at a 2-month follow up.

### Results

Sixty-one NDs met eligibility criteria. Eighty-nine percent of participants agreed or strongly agreed that they were satisfied with the course. There was a significant increase in self-reported skill and objectively measured EBP knowledge, but no substantive change in EBP attitudes or use of evidence over time. Some participants indicated the level of difficulty was too high while others reported that it could have been more difficult. Participants also wanted more opportunities to practice the skills being taught in the course.

**Data availability statement:** The data collected for this project are sensitive and may allow for identification of participants. The study participants did not provide consented for the publication of raw data. Requests for access to the data are to be sent to the Chair of the Research Ethics Board of the Canadian College of Naturopathic Medicine (REBChair@ccnm. edu).

**Funding:** This project was awarded a competitive grant from the Canadian CAM Research Fund (CCRF). The funders had no role in study design, data collection and analysis, decision to publish, or preparation of the manuscript.

**Competing interests:** No authors have competing interests.

## Conclusions

Delivery of the co-designed EBP CE course was found to be both feasible and acceptable. Preliminary evidence suggests that participation in the course was associated with improvements in EBP knowledge and skill. Participants provided actionable suggestions to improve the course in future iterations.

## Introduction

Evidence-based practice (EBP) involves the conscientious use of the best available evidence, in combination with clinician experience and patient preference in an effort to optimize clinical care [1]. It involves the systematic, conscientious, and judicious analysis and application of published research to clinical decision making [2]. The development of evidence-based healthcare is attributed to the work of several individuals including Archie Cochrane [3], McMaster University investigators [2] and Sacket et al [1]. A number of benefits have been attributed to the use of EBP, including improved patient outcomes and reduced health care costs [4,5]. Accordingly, there has been increased attention to integrate EBP in a wide range of healthcare disciplines, including traditional complementary and integrative medicine (TCIM) systems such as naturopathic medicine. While studies have evaluated the effects of EBP training in medical education and continuing education [6,7], evaluation of the effects of EBP training in naturopathic medicine has not been undertaken previously [8].

Naturopathic Medicine is a distinct system of healthcare that combines scientific and traditional evidence to support the delivery of a range of natural therapeutic approaches [9]. The practice is defined and guided by a set of principles [10]. The therapeutic modalities used by naturopathic doctors vary by jurisdiction but often include dietary counselling, nutritional supplementation, herbal medicine, homeopathy, physical medicine, and lifestyle counselling [9]. In order to qualify for licensing, Naturopathic Doctors (NDs) are required to complete a bachelor's degree, a 4-year naturopathic medicine program at an accredited college and successfully complete licensing exams. There are approximately 2400 NDs in Canada [11]. Naturopathic medicine is licensed in British Columbia, Alberta, Saskatchewan, Manitoba, Ontario and Nova Scotia. The naturopathic medicine programs currently include a substantial focus on evidence-based practice; however, this represents a significant increase compared to previous cohorts. In 2002 a series of papers was published that presented an approach to teaching EBP for TCIM [12–15]. The National Institutes of Health National Center for Complementary and Alternative Medicine funded nine R25 education grants from 2005 to 2011 for the purpose of designing increased quantity and quality of EBP curriculum content at institutions educating TCIM clinicians [16].

Several authorities have argued that naturopathic practice is opposed to EBP [17,18]; however, this suggestion is not adequately supported by current evidence. Qualitative research has revealed primarily favourable views of EBP

among practitioners of Naturopathic Medicine [19]. Similarly, a recent survey of Canadian NDs reported that attitudes, use and skills in EBP were all relatively high, noting that there was room for improvement [20]. Additionally, 93% of respondents reported an interest in undertaking further education to improve EBP skills. The findings of this survey highlighted an opportunity to improve EBP uptake among practicing Canadian NDs by increasing EBP skills acquisition. Prior research conducted in undergraduate naturopathic students suggested that both sample size determination, and program design were important considerations for educational interventions aimed at developing EBP skills [16,21].

Recognising the potential value of EBP education, a process was undertaken to co-designed an EBP Continuing Education (CE) course that was tailored to the needs of Canadian NDs [22]. Co-design involves the collaboration of researchers and end-users in an effort to conduct research that is relevant and useful [23]. Practicing NDs, including representation from educators, leaders in the profession, and new graduates, provided input into course design via three focus groups [24]. The findings from the focus groups were combined with best practices in EBP education [25,26] to create a CE course. Overall, the focus group participants indicated a high level of interest in participating in the course and an interest in learning foundational EBP skills such as searching for evidence, critically appraising evidence, assessing risk of bias and applying evidence to clinical care. Additionally, some participants were interested in exploring these topics in a way that was tailored to naturopathic medicine, taking into consideration the philosophy and principles of naturopathic medicine, and valuing a range of sources of evidence [22].

The Theory of Planned Behaviour was also used to inform the design of the course [27]. The course attempted to increase perceived behavioural control through the use of skill building exercises. Group discussions about the benefits of EBP, and guest lectures from leaders within and beyond the naturopathic profession were also undertaken to facilitate changes in subjective norms and attitudes.

The present publication reports on the delivery and evaluation of the abovementioned CE course [22]. The primary objective of the project was to assess the feasibility and acceptability of the pilot EBP CE course for Canadian NDs. Secondary objectives were to assess changes in self-reported EBP attitudes, skills and use, and objectively measured EBP knowledge. It is anticipated that these findings will be used to inform future EBP training opportunities for NDs.

## Methods

### Study design

Evaluation of the EBP CE course was undertaken using a pre-post design with a 2-month follow up after course completion.

### Continuing education course

The course consisted of five, one-hour, weekly sessions that were delivered synchronously through a virtual platform during the months of November and December, 2021. The course was delivered by the continuing education department of the Canadian College of Naturopathic Medicine; this was the first time an EBP course was delivered. Sessions were video recorded with transcription, which were available to participants for viewing following each session. The topics covered in each session are listed in Table 1. The sessions included a combination of didactic teaching, group discussion, and skill-building exercises. When co-designing the course, participants recognized the wide range of EBP skills and knowledge within the profession and suggested having material tailored to those needing to review basic concepts and those wanting to learn more advanced topics; this material was incorporated into the first and final sessions of the course. The sessions were primarily facilitated by MA with segments of instruction from ML and KC and a professor from the methodology department of a local university. Participants received a digital handout summarizing important information for future reference, including links to important resources and tools (supplemental file 1).

**Table 1. Evidence Based Practice Continuing Education Course Topics.**

| Session | Title | Topics Covered |
|---|---|---|
| 1 | The Basics of Reading Scientific Research | - Overview of Evidence Based Practice: history, purpose, definition, strengths and limitations<br>- Review of the different types of study methodologies (clinical trials, cohort, case-control and cross-sectional studies, secondary research), including key elements of each, strengths and limitations |
| 2 | Finding Scientific Evidence | - Review the different sources of evidence that can be used to inform clinical practice<br>- Development of a structured, answerable clinical question (PICO)<br>- Strategies for developing an efficient search strategy using terms and limits<br>- Overview of how to navigate commonly used databases<br>- Exercise where participants practiced designing and executing a search strategy; involved an iterative process and group discussion of strategies |
| 3 | Appraising Scientific Evidence | - Overview of appraisal: purpose, rationale, methods<br>- Approaches for interpreting different types of studies<br>- Definition of bias including the types of bias and potential impact on study results<br>- Review of basic statistics needed to interpret studies<br>- Exercise where participants practiced critically appraising a clinical trial; included group discussion |
| 4 | Applying Scientific Evidence | - Comparison of different types of study outcomes<br>- Translating evidence, implementing evidence<br>- How to manage uncertainty in clinical decision making (lack of evidence, conflicting findings)<br>- Discussion of strategies for viewing research in the context of the naturopathic principle and philosophy<br>- Discussion of strategies for integrating scientific evidence with other sources of evidence<br>- Group discussion about the role and application of different sources of evidence in the practice of naturo-pathic medicine |
| 5 | Advanced Topics | - Advanced critical appraisal topics<br>- Why and how to write a case report |

## Study participants

Criteria for participation were as follows:

1) ND with an active licence to practice in one of the following Canadian provinces: British Columbia, Alberta, Saskatchewan, Manitoba, Ontario or Nova Scotia,

2) willingness to provide informed written consent, and

3) ability to communicate in English.

## Sample size

This study was designed to assess feasibility and acceptability as primary outcomes. Because this project is the first to assess changes in EBP skills, attitudes and use following EBP training among NDs, the magnitude of change was unknown, preventing formal sample size calculation. Based on Sim and Lewis's rule of thumb for pilot feasibility studies, the target sample size was determined to be at least 55 participants [28].

## Outcomes

The primary outcomes (i.e., feasibility and acceptability) were assessed quantitatively and qualitatively. Feasibility was assessed by counting (a) the number of participants recruited within the 3-week recruitment period, (b) the number of participants attending each of the CE sessions, and (c) the number of participants completing the data collection instruments. In assessing attendance, participants were counted as having attended a session if they were present for at least some part of the live delivery (i.e., visual proof of attendance) or if they watched the recording of the session through the online platform used to deliver the course (i.e., using user statistics).

To assess acceptability of the program materials and delivery, a satisfaction survey was administered at the end of the course. The following questions were included:

1. I was satisfied with the EBP CE Course: Strongly disagree, Disagree, Agree, Strongly Agree

2. I would recommend this EBP CE course to my colleagues: Strongly disagree, Disagree, Agree, Strongly Agree

3. The part of the course that I found most beneficial was: (open text response)

4. The part of the course that I found least beneficial was: (open text response)

5. One change that I would make to improve the course would be: (open text response)

Acceptable course satisfaction was defined as 75% of participants 'agreeing' or 'strongly agreeing' to the question "I was satisfied with the EBP CE Course".

Secondary outcomes were measured using the Evidence-Based Practice Attitudes and Utilization Survey (EBASE) [29]. This self-administered questionnaire is used to assess six constructs including use of EBP, EBP skills, EBP attitudes, EBP-related training, and barriers and enablers to EBP uptake, as well as demographic characteristics. Because the EBASE was originally developed for TCIM practitioners in Australia, minor modifications to a small number of demographic questions were made to reflect regionally specific terminology without altering the meaning of any survey items.

For the purpose of the present project, only questions related to use, skills, attitudes and demographics were asked in order to minimize participant burden while capturing the data most relevant to the study objectives. Data on the former three constructs were used to generate sub-scores related to EBP use, skills, and attitudes. The psychometric properties of EBASE have been established, and the instrument has demonstrated acceptable content validity, internal consistency and test-retest reliability [29].

EBASE has been used in cross-sectional studies involving a a wide range of health care disciplines, including nursing students [30], chiropractors [24], osteopaths [31], Western herbalists [32], yoga therapists and traditional Chinese medicine practitioners [32], from across several geographic regions, including North America [20], Europe [31] and Australia [32], allowing for comparisons between professions and regions. EBASE has been used previously to assess changes in use, skills, and attitude following exposure to an EBP educational program, but only in chiropractic and nursing samples [29,30,33].

While EBASE does assess self-reported EBP skill level, we acknowledge that perceived EBP knowledge and skills may be prone to bias. As such, we administered the EBASE in combination with a 16-question, multiple-choice quiz at baseline and after the course in order to objectively assess changes in EBP knowledge. This quiz was created specifically for this research and included questions about research designs, interpretation of basic statistics, designing a literature search strategy, and assessing study quality.

## Recruitment

The CE course was offered through the Canadian College of Naturopathic Medicine (CCNM) CE department and advertised through a post on the CCNM CE Website, emails to CCNM alumni and faculty members, posts on CCNM social media platforms, and through the CCNM alumni newsletter (noting that a large number of Canadian NDs are alumni of CCNM). An invitation to participate in the course was also included in the Ontario Association of Naturopathic Doctors monthly classified email. Additionally, social media posts were created for naturopathic virtual communities of practice that include Canadian members. All individuals who responded to advertisements for the co-design phase of the study were also invited to participate in the current study via email. Snowballing by 'word of mouth' was encouraged.

The recruitment strategy was designed to be multi-faceted to ensure all eligible licensed Canadian NDs had an opportunity to complete the course. Participation in the course was voluntary and participants self-selected to participate. An incentive was offered to encourage participation and retention. Participants who completed the data collection instruments at all time points were entered into a draw for a prize (i.e., one of two CAD $500 cash prizes).

### Data collection

All data were collected through the institution's secure electronic data capture system, Research Electronic Data Capture (REDCap) [34]. Participants received up to four automated email invitations at each data collection time point. The first invitation (sent prior to course commencement) invited participants to complete an eligibility screening questionnaire and to review and sign the informed consent form. Eligible participants were then invited to complete the baseline data collection form (week 0). This included completion of EBASE and the EBP knowledge quiz. Following the final session (week 5), participants were invited to complete EBASE and the EBP knowledge quiz a second time, as well as the course satisfaction survey. Two months after the final session (February 2022, week 13), participants were invited to complete the use sub-scale of EBASE a third time.

### Data analysis

Data were imported into Microsoft Excel for cleaning and coding, and subsequently into R for analysis [35]. In cases of missing data, the last observation was carried forward. Data from participants who did not consent to participate or who did not complete the baseline assessment were removed from all analyses.

Categorical demographic data were reported using frequencies and percentages. The *a priori* statistical plan was to use descriptive statistics for demographic data only (due to the pilot nature of our study). Changes in the secondary outcomes (EBASE sub-scores and knowledge quiz) were analyzed using paired sample t-tests (t) if data were normally distributed or the Wilcoxon signed rank test (W) if data were not normally distributed. Normality was assessed using the Shapiro-Wilk test. The effects were considered statistically significant if the p-value for statistical tests was less than 0.05.

Exploratory analyses assessed for differences in demographic variables (age, length of time in clinical practice, time spent on research and teaching), course attendance, and baseline quiz score. Kendall's Tau Correlation Coefficient ($\tau$) was used to assess for correlations between ordinal variables. A coefficient between 0.10–0.29 was defined as a weak association, while a moderate association was defined as 0.30–0.49 and a strong association was defined as 0.50–1.00 [36]. Qualitative data from the satisfaction survey were analyzed using structured framework analysis, an approach commonly used in evaluation studies [37]. The purpose of the satisfaction survey was to inform revision of the course in subsequent iterations. For each element of the course (e.g., content, format), we identified data related to what the participants found helpful, unhelpful, and suggestions for improvement.

### Ethical approval

This study received approval from the Research Ethics Board of the CCNM (CCNMREB038.Aucoin.Newton.Cooley). All participants provided informed consent to participate.

### Results

Course registration opened on October 19 2021, and closed November 5 2021. Eighty-one participants registered for the course (Fig 1), with 61 participants meeting criteria for the study, agreeing to participate, and completing baseline assessments (week 0). Fifty-seven participants completed the post-course (week 5) and follow-up assessment (week 13).

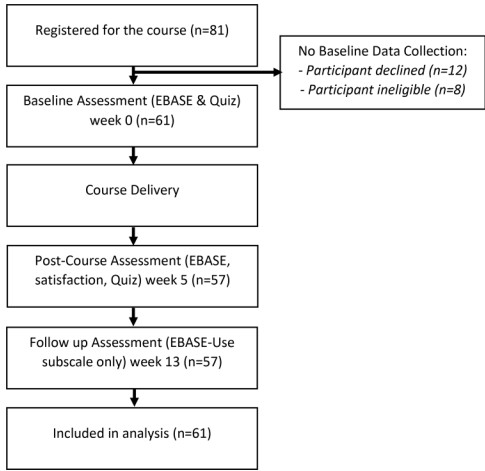

**Fig 1. Flow of participants through the study.**

## Demographics

The demographic characteristics of participants that completed the baseline assessments are presented in Table 2. Participants were primarily aged between 30 and 49 years (75%), and two-thirds (64%) identified as women. There was variety in the length of time since participants completed their highest qualification and the clinical setting in which participants practiced. The majority of participants lived in Ontario (74%).

## Attendance

Of the 61 participants included in the analysis, 39 participants (64%) attended four or five of the sessions in person or had viewed the recording (Fig 2). Thirteen participants (21%) attended one or zero sessions.

## Satisfaction survey

Participants expressed a high degree of satisfaction with the course. Eighty-nine percent of participants "agreed" or "strongly agreed" that they were satisfied with the course. Similar results were found for the number of participants who would recommend the course (Table 3).

Participants provided a range of comments about aspects of the course that were most and least helpful, as well as suggestions for improvement. There were a large number of positive comments about the course as a whole ("This course was a great comprehensive review of evidence-based practice"; "I found all of it beneficial"; "This was an excellent course, I have no suggestions to improve"). The main themes that emerged related to the level of difficulty, course content, course format and resources (Table 4).

With respect to the level of difficulty of the course, a wide range of responses were received. Some respondents commented that the review of the basic concepts of EBP was helpful ("Refreshing my past knowledge on the types of trials, p-value and odds ratio and how to search better"), while others cited this material as unhelpful because it was too high-level or simple ("intro lecture - mostly review though I understand this is important to establish a baseline"; "I would love to dive deeper still but I can also appreciate that this is targeted to varying levels of comfort"). Similarly, some participants appreciated the more advanced topics while others reported that some sections were too challenging and that the pace of the course was too fast. Many participants suggested offering the course at different levels of difficulty (i.e., Basic, advanced) in the future.

Table 2. Demographic characteristics of participants (n = 61).

| Characteristic | Frequency, n (%) |
|---|---|
| **Age** | |
| *20-29 years* | 3 (5) |
| *30-39 years* | 24 (39) |
| *40-49 years* | 22 (36) |
| *50-59 years* | 6 (10) |
| *60-69 years* | 3 (5) |
| *70 + years* | 3 (5) |
| **Gender** | |
| *Women* | 39 (64) |
| *Men* | 20 (33) |
| *Other gender identities* | 2 (3) |
| **Highest qualification** | |
| *Naturopathic Doctor* | 53 (93) |
| *Master's degree* | 8 (7) |
| **Years since completing highest qualification** | |
| *< 1 year* | 1 (2) |
| *1-5 years* | 16 (26) |
| *6-10 years* | 17 (28) |
| *11-15 years* | 7 (6) |
| *16 + years* | 20 (16) |
| **Hours per week in clinical (naturopathic medicine) practice** | |
| *0 hours* | 1 (2) |
| *1-15 hours* | 28 (39) |
| *16-30 hours* | 25 (20) |
| *31-45 hours* | 6 (5) |
| *46 + hours* | 1 (1) |
| **Hours per week participating in research** | |
| *0 hours* | 21 (34) |
| *1-15 hours* | 40 (66) |
| *16-30 hours* | 0 (0) |
| *31-45 hours* | 0 (0) |
| *46 + hours* | 0 (0) |
| **Hours per week teaching in the higher education sector** | |
| *0 hours* | 24 (39) |
| *1-15 hours* | 26 (43) |
| *16-30 hours* | 8 (7) |
| *31-45 hours* | 3 (2) |
| *46 + hours* | 0 (0) |
| **Clinical setting in which naturopathic medicine is predominantly practiced** | |
| *Solo practice* | 22 (36) |
| *With a combination of complimentary and conventional medical health practitioners* | 16 (26) |
| *Within a clinical institution (e.g., Hospital, nursing home)* | 14 (23) |
| *With a group of conventional medical/allied providers* | 9 (15) |
| *Within an educational institution* | 0 (0) |

*(Continued)*

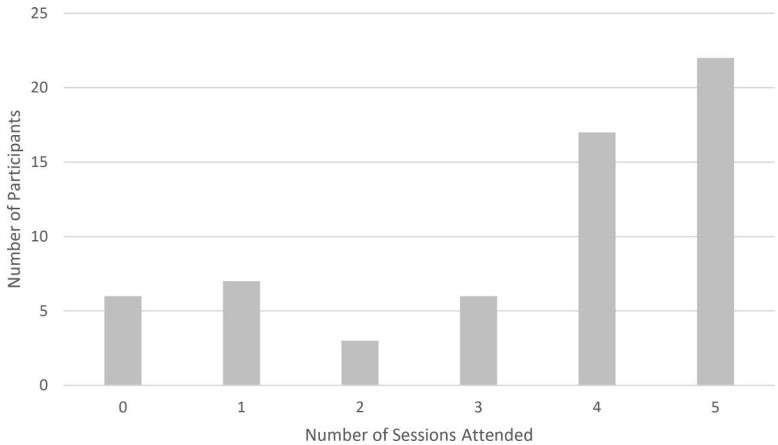

**Table 2.** (Continued)

| Characteristic | Frequency, n (%) |
|---|---|
| **Geographical location** | |
| *Ontario* | 45 (74) |
| *British Columbia* | 10 (16) |
| *Alberta* | 2 (3) |
| *Manitoba* | 2 (3) |
| *Quebec* | 1 (2) |
| *Saskatchewan* | 1 (2) |
| *Nova Scotia* | 0 (0) |
| *Newfoundland* | 0 (0) |
| *New Brunswick* | 0 (0) |
| *Prince Edward Island* | 0 (0) |
| *Northwest Territories* | 0 (0) |
| *Yukon* | 0 (0) |
| *Nunavut* | 0 (0) |

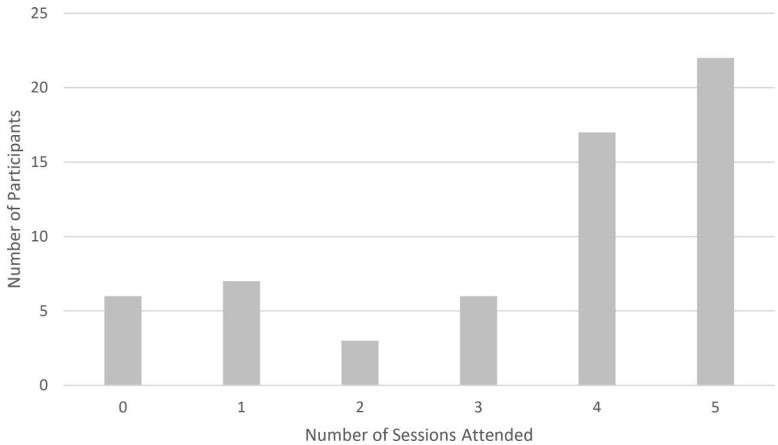

**Fig 2. Participant attendance among those participants included in analysis (n = 61).**

Participants appreciated many components of the course content, including sections on critical appraisal, searching for evidence and statistics. There were both positive and negative comments about the inclusion of material related to applying research to the practice of naturopathic medicine The suggestions for improvement included an interest in learning more about statistics, including clinical cases, and more practical activities.

With respect to the course format and resources, participants reported a combination of positive and negative comments about the online delivery format. There was a high level of appreciation for the interactive and practical components of the course and an interest in providing more opportunities to practice the skills being taught ("More practice in reviewing studies [and] searching for literature"). Participants suggested reviewing more sample journal articles and patient cases. Several participants were interested in undertaking quizzes to assess their own comprehension of the material and some suggested homework or assignments ("Make it longer with more assignments and practice."). There were also many calls for making the course longer and more in-depth ("More time!!").

While not part of the formal evaluation, some of the comments posted on the webinar platform were very insightful. In response to the discussion about the role of evidence in naturopathic medicine, one participant posted the comment "I

**Table 3. Satisfaction questionnaire results following course participation (week 5).**

| Prompt | Strongly disagree n (%) | Disagree n (%) | Agree n (%) | Strongly Agree n (%) |
|---|---|---|---|---|
| I was satisfied with the course | 5 (8.8%) | 1 (1.8%) | 27 (47.4%) | 24 (42.1%) |
| I would recommend the course | 5 (8.8%) | 1 (1.8%) | 25 (43.9%) | 26 (45.6%) |

**Table 4. Themes identified in response to questions about the most helpful and least helpful aspects of the course and opportunities for improvement.**

| Themes | Most Helpful | Least Helpful | Suggestions for Improvement |
|---|---|---|---|
| Level of Difficulty | - basics lecture<br>- advanced lecture | - basics lecture (too easy, too much review)<br>- advanced lecture (too difficult) | - basic/intermediate/ advanced courses<br>- more content, more in depth course |
| Content | - critical appraisal<br>- searching for evidence<br>- risk of bias<br>- statistics<br>- application to naturopathic medicine | - application to naturopathic medicine | - more statistics<br>- more practice searching<br>- more cases and example papers<br>- more content about getting involved in research |
| Format | - interactive components<br>- practical exercises<br>- modules<br>- online delivery<br>- guest lecturers | - timing (too late in the evening)<br>- pace (too fast)<br>- online delivery<br>- guest lecturers | - earlier time<br>- longer sessions<br>- more time for questions<br>- modules more spaced out<br>- increase time spent on opportunities to practice skills |
| Resources | *(no data elicited)* | *(no data elicited)* | - quiz for each module to review and assess comprehension<br>- assignments and homework |

think our profession overly depends on clinical pearls and anecdotal info. I hope we can shift this…courses like this are a good start! Thank you." Overall, the comments communicate a high level of interest in the course and material, a high degree of interest in developing EBP skills, and a need for further EBP education opportunities within the Canadian naturopathic profession.

### EBP knowledge

The average score for the EBP knowledge quiz was 10.8 (IQR 9,13; out of a maximum score of 16) at baseline and 12.1 (IQR 11,13) after the course. The change in quiz score from pre-course to post-course was statistically significant ($W = 168$, $p < 0.001$). The number of participants with a score of 10 or less decreased from 21 (34.4%) to 4 (6.6%) between the two assessment timepoints (week 0 and week 5). The distribution of quiz scores before and after the course are displayed in Fig 3, noting that data were not normally distributed.

### Self-rated EBP attitudes, skill, and use

Mean EBASE sub-scale scores at each timepoint are displayed in Table 5. At baseline, the mean Attitude sub-score of 32.7 reflected predominantly agree to strongly agree responses for most questions (range of values for subscale: 8–40). There was no statistically significant change in mean Attitude sub-scores between week 0 and week 5 ($t(60) = 0.1023$, $p > 0.05$). The baseline Skill sub-score of 38.9 reflected an average to somewhat advanced skill level (range of values for sub-score: 13–65). The skills sub-score increased significantly between week 0 and week 5 ($t(60) = 5.483$, $p < 0.001$). The week 5 (post-course) mean Skill sub-score was consistent with a somewhat advanced to advanced

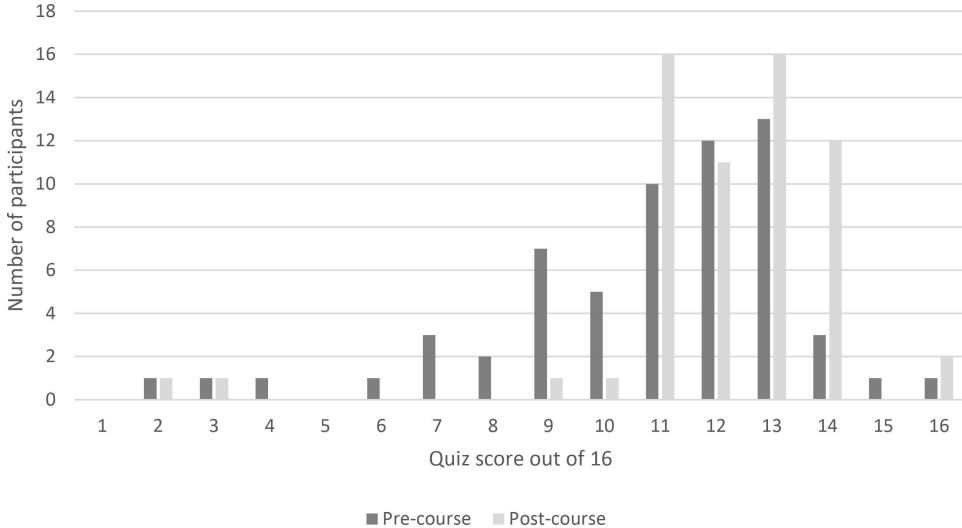

**Fig 3. Distribution of participant scores on the evidence-based practice quiz completed before and after course participation.**

**Table 5. EBASE Sub-scores, before and after course participation (n = 61).**

|  | Pre-course score (week 0), mean (IQR) | Post-course score (week 5), mean (IQR) | 2- month follow-Up, mean (IQR) | P value |
|---|---|---|---|---|
| EBP Attitude Sub-score | 32.7 (30, 36) | 32.7 (30, 36) | Not assessed | p = 0.919 |
| EBP Skills Sub-score | 38.9 (32, 45) | 43.0 (37, 50) | Not assessed | p < 0.001 |
| EBP Use Sub-score | 20.7 (14, 28) | 20.5 (15, 24) | 19.3 (14, 27) | p = 0.855 |

level of skill. Attitude and Skill subscales data were normally distributed. Mean Use sub-score at baseline was 20.7, which was consistent with the highest quartile for this scale (range of values for subscale: 0–24). Mean Use sub-scores were not significantly different (p > 0.05) between weeks 0, 5 and 13. Use subscale data were not normally distributed.

Participants were asked about the percentage of their clinical practice that was based on clinical research evidence. The majority of participants (66%) responded with a moderate or large proportion at baseline, which increased to 72% of participants at week 5 (post-course) and 75% of participants at 2 months follow-up (Fig 4).

## Exploratory analyses

Exploratory analyses identified a number of significant correlations between participant characteristics and study outcomes at week 5. A weak positive association was observed between participant age and change in attitude score (τ = 0.26, p = 0.021). Time since graduation and hours spent in clinical practice, research and teaching were not correlated with changes in any outcome.

Baseline use of evidence was moderately inversely correlated with change in evidence use score (τ = −0.367, p < 0.001) and baseline quiz score was moderately inversely correlated with change in knowledge quiz score (τ = −0.371, p < 0.001). There was also a weak positive association between course attendance rate and change in quiz score (τ = 0.228, p = 0.041) and change in self-reported skill (τ = 0.211, p = 0.051).

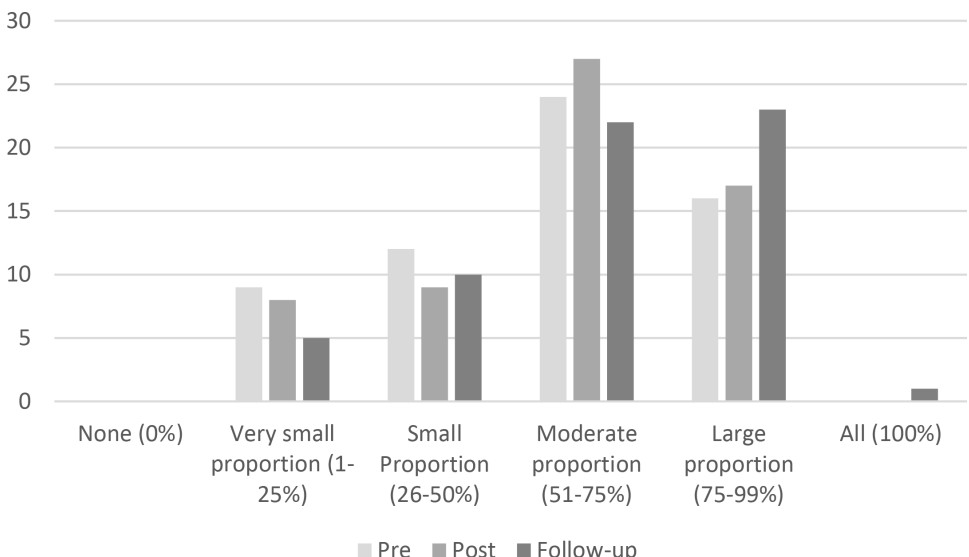

**Fig 4. Percentage of clinical practice based on evidence from clinical research, as self-reported by participants (n = 61).**

## Discussion

This is the first known study to evaluate a co-designed EBP CE course for Canadian NDs. The EBP CE course was found to be feasible and acceptable to Canadian NDs, as evidenced by an adequate level of course enrollment, course attendance, questionnaire completion, and participant satisfaction. Completion of the course was shown to be associated with improvements in EBP knowledge scores and self-reported skill levels; however, EBP attitudes and use of evidence did not change significantly over the study period. Participants offered many suggestions to improve the course, which will be incorporated into future iterations of the course.

### Feasibility and acceptability

The study findings support the feasibility of the EBP CE course for Canadian NDs. Although the number of participants was small relative to the number of Canadian NDs, the study was successful at enrolling the target number of participants within a short (3-week) period of time. Participants' willingness to enroll in this course may have been related to a number of considerations, one being the co-design process, which aimed to tailor the course to the needs of Canadian NDs. Approval of the course for CE credits, the relatively low cost of the course (compared to other CE offerings for Canadian NDs, CAD $97), and the provision of a participation incentive (e.g., awarding of prizes to two participants who complete the questionnaires) may also explain the high willingness of participants to enrol in the course. It is noted that the majority of participant were from Ontario and British Columbia; these are the provinces in which CCNM has its two campuses and the number of licensed NDs is highest. The low number of participants from Quebec, New Brunswick, Prince Edward Island, Newfoundland or the Canadian Territories is likely related to the fact that NDs were not eligible for licensing in these jurisdictions at the time the course was delivered.

The participant satisfaction questionnaire provided further insight into the acceptability of the CE course. In fact, the level of satisfaction exceeded our pre-determined criteria for acceptability. Similarly, the written responses from participants suggested a high level of satisfaction with the course content and delivery as well as an interest in learning EBP. Three participants from the focus group co-design phase participated in the evaluation of the course; as 61 participants were included in the analysis, it is unlikely that the level of satisfaction was inflated by participation from these individuals.

Overall, this suggests that Canadian NDs are supportive of EBP training; however, it is acknowledged that this interest was expressed by a group of individuals who self-selected to participate in the course.

The findings of this project yielded information that could be considered by others designing EBP courses for TCIM practitioners or in other jurisdictions. Participants expressed a high level of appreciation for the interactive and practical components of the course. There was also considerable interest in additional opportunities to practice the skills being developed through the course (e.g., by including more sample articles and cases). While the setting of the course precluded involvement of actual patient cases, it is possible that simulated cases or other case-based approaches may have enhanced learning. Indeed, previous research has shown clinically integrated teaching to be more effective in improving EBP skills, knowledge, skills and attitudes relative to standalone teaching [6]. There was also interest among participants to undertake quizzes, assignments and other homework to check their comprehension, and consolidate their understanding of key concepts. These comments suggest a high level of interest in engaging meaningfully with EBP training, and a genuine interest in the development and refinement of useable skills.

Participant feedback highlighted the diverse range of EBP educational needs of Canadian NDs. Despite embedding foundational and advanced content within the present course, many participants found the course either too challenging or not sufficiently challenging. This suggests that subsequent courses should be tailored to a basic or advanced level to better accommodate the range of abilities among members of the profession.

### Use of evidence

Our findings suggest that participation in the EBP course had little impact on a participants' use of evidence. Notwithstanding, participants did report a high level of evidence use at baseline, which was significantly higher than EBASE Use sub-scores reported in a recent cross-sectional survey of Canadian NDs(20). Given the high level of evidence use at baseline, a ceiling effect may have been present, which may explain the negligible change in use score over time. Indeed, we found a significant inverse correlation between baseline use of evidence and change in use over the course. In other words, participants using evidence infrequently prior to course participation were more likely to increase their use as a result of participation, while those reporting higher baseline use were more likely to maintain their level of use. Data collected during the CE course co-design focus-group project further supports this proposition [22], with many participants indicating they were satisfied with the amount of evidence they were using in practice, but wanted to improve the efficiency of their searches and ability to critically appraise studies.

The negligible change in EBP use observed in the current study can be further understood by comparing findings with a 2016 study evaluating a 16-week (157-hour) mandatory EBP course for Australian nursing students [30]. While changes in Attitudes and Skill sub-scores were comparable between studies, Use sub-score in the cohort of nursing students increased on completion of the course. This may be because the Use sub-score in nursing students was low at baseline (i.e., score of 7) compared to that reported among Canadian NDs in the present study (i.e., score of 20). This study showed that in a sample of students who were not self-selected, lower levels of use at baseline were modified following exposure to an EBP course. Overall, the limited change in EBP use in the present study could be attributed to the enrollment of self-selected participants who were using evidence at a relatively high level prior to the course.

### Evidence based practice attitudes

Similar to Use sub-scores, participant Attitude sub-scores in the current study were also relatively high at baseline; and likewise, attitude sub-scores did not change significantly over time. This might suggest that individuals interested in participating in an EBP course already possess favourable attitudes towards EBP. It is also possible that the educational intervention may not have been of sufficient intensity or duration to shape participant attitudes, noting that for many healthcare disciplines, there is limited evidence to suggest that Evidence Based Medicine (EBM) training is able to modify EBM attitudes [38]. That said, a systematic review assessing the effects of postgraduate EBM training found that interventions

involving standalone teaching were unlikely to change participant attitudes, whereas interventions integrating training into clinical practice were more likely to result in attitudinal change [4]. This finding is consistent with participant suggestions to include more case examples in the course. Overall, the lack of change in attitudes may be attributed to the predominantly positive views at baseline (i.e., a ceiling effect), or quite possibly, may reflect the inherent difficulties in shifting practitioner attitudes in favour of EBP.

### Evidence based practice skill and knowledge

The results of our study suggest that participation in the course resulted in an increase in EBP skills and knowledge; however, several factors appear to have impacted the degree of improvement between participants. Changes in skill sub-scores and knowledge quiz scores were both positively correlated with course attendance suggesting that improvements in these scores were the result of course participation. A significant inverse correlation was observed between baseline quiz score and change in quiz score, as well as baseline EBASE Skill sub-score and change in EBASE Skill sub-score. Participants with lower baseline knowledge and skill were able to increase their knowledge to a greater degree, compared to those with high skill and knowledge at baseline. The smaller increase in knowledge and skill among those with high baseline scores may be related to a ceiling effect or it may suggest that the course was better suited to addressing the knowledge needs of those with lower baseline levels. This provides further justification for the need to deliver the CE course with multiple levels of difficulty. It is acknowledged that the mean baseline score of 10.6 out of 16 and negative skew of the data suggest that a more difficult quiz may have been required to detect changes in the knowledge of more advanced participants. Overall, there is preliminary evidence to suggest that course participation was associated with an increase in EBP knowledge and skill.

### Strengths and limitations

A strength of the present study was the co-designed intervention. The use of co-design increased the likelihood of producing a course that would be acceptable to, and useful for the target audience, and thus would have greater impact; which is consistent with accepted knowledge mobilization principles [39]. Further, the course was evaluated by a modest sample of clinicians, and the outcomes were assessed using a widely used and validated survey. We also mitigated the risk of overestimating the effectiveness of the course by ensuring data from all participants were included in the analysis (i.e., by using the last-observation-carried-forward approach for any missing data). Lastly, the demographics of the present sample were similar to the demographics of the participants in the previous Canadian survey [20], allowing for comparison.

A notable limitation of this study was the absence of a control group. Although it is likely that changes in EBP skills and knowledge were the result of participation in the course, this can not be confirmed. The association between increased attendance and higher change in knowledge score suggests that course attendance may have been responsible for this improvement. Although responses to questionnaires were anonymous, participant bias cannot be discounted. Another limitation of the study was that participants were self-selected, meaning those with less favourable EBP attitudes, lower EBP skills or lower evidence use may not have taken part in the study. Thus, the impact of the course on these non- participating NDs remains unclear. Outcome assessment was from the participant perspective only; future research on this topic may benefit from evaluating the course from a variety of perspectives, although notable differences between evaluation and research exist [40]. Finally, our study is limited by single site, instructor-effects, and the period of follow-up assessment. Knowledge retention, skills or attitudes may shift with variations in those factors.

### Importance

The findings of this study suggest that training in EBP is of interest to Canadian NDs. Given that the EBP educational content in the naturopathic curriculum has increased over time, those practitioners who graduated earlier, received significantly less training in this subject. Within all healthcare fields, there is a movement toward increasing use of EBP and the

naturopathic profession is no exception. There is need, interest and opportunity for more CE courses on EBP to be offered to Canadian NDs, international NDs and members of other CIH professions that could use a similar approach to the one presented in this manuscript.

## Conclusions

Delivery of a co-designed EBP CE course to Canadian NDs was shown to be both feasible and acceptable. Preliminary evidence suggests that participation in the course was associated with improvements in EBP knowledge and skill. Participants provided actionable suggestions to improve the course in future iterations such as delivering the course with multiple levels of difficulty. Future research could be conducted to evaluate innovations in delivery, modified course design or content, different populations or geographic settings, or larger participant pools.

## Supporting information

**Supplemental file 1. EBP course resources and tools.**
(PDF)

**Acknowledgments**: The authors wish to acknowledge contributions from Dr. William Bettger (analysis feedback), Rebecca Lester (graphic design assistance), and Emma Di Paolo (data management). Course delivery was facilitated with support from members of the Canadian College of Naturopathic Medicine Continuing Education Department, particularly Dr. Sasha Tahiliani, ND.

## Author contributions

**Conceptualization:** Monique Aucoin, Genevieve Newton, Kieran Cooley.

**Formal analysis:** Monique Aucoin, Genevieve Newton, Matthew Leach, Kieran Cooley.

**Funding acquisition:** Kieran Cooley.

**Methodology:** Monique Aucoin, Genevieve Newton, Matthew Leach, Kieran Cooley.

**Project administration:** Monique Aucoin.

**Supervision:** Genevieve Newton, Matthew Leach, Kieran Cooley.

**Writing – original draft:** Monique Aucoin.

**Writing – review & editing:** Genevieve Newton, Matthew Leach, Kieran Cooley.

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
