## [Decision Letter · Decision Letter 0]

14 Mar 2025

PONE-D-25-02800Evaluation of an Evidence-Based Practice Continuing Education Course for Canadian Naturopathic DoctorsPLOS ONE

Dear Dr. Cooley,

Thank you for submitting your manuscript to PLOS ONE. After careful consideration, we feel that it has merit but does not fully meet PLOS ONE’s publication criteria as it currently stands. Therefore, we invite you to submit a revised version of the manuscript that addresses the points raised during the review process.

We look forward to receiving your revised manuscript.

Kind regards,

Jenny Wilkinson, PhD

Academic Editor

PLOS ONE

Journal Requirements:

This project was awarded a competitive grant from the Canadian CAM Research Fund (CCRF).  

3. In the online submission form, you indicated that data cannot be shared publicly because this was not approved by the research ethics board or included in the participant informed consent form. Data requests can be made to the corresponding author maucoin@ccnm.edu and will be reviewer by the Research Ethics Board. 

Reviewers' comments:

Reviewer's Responses to Questions

**Comments to the Author**

1. Is the manuscript technically sound, and do the data support the conclusions?

Reviewer #1: Yes

Reviewer #2: Yes

Reviewer #3: Yes

2. Has the statistical analysis been performed appropriately and rigorously? 

Reviewer #1: Yes

Reviewer #2: Yes

Reviewer #3: Yes

3. Have the authors made all data underlying the findings in their manuscript fully available?

Reviewer #1: Yes

Reviewer #2: Yes

Reviewer #3: No

4. Is the manuscript presented in an intelligible fashion and written in standard English?

Reviewer #1: Yes

Reviewer #2: Yes

Reviewer #3: Yes

5. Review Comments to the Author

Reviewer #1: PONE-D-25-02800 "Evaluation of an Evidence-Based Practice Continuing Education Course for Canadian Naturopathic Doctors".

Comments to the Authors

Thank you for your work on this manuscript. This research will add to the literature in a number of different areas. The manuscript is well-written and meets the journal’s publication criteria.

Originality

This is an original piece of work that can add to the literature in the context of designing courses that focus on evidence-based practice (EBP) in the context of continuing education for naturopathic doctors.

Introduction

Relevant background information and a rationale for the study are provided. Study aims are clearly stated.

Methods

The Methods section is clearly organised and provides relevant information under each section. Study measures are clearly described. Detailed information of the analysis process is provided.

Results

The results are clearly outlined and answer the research aims. The Tables and Figure support the results.

Discussion

The main findings of the study are discussed, and some links are made to earlier research that is applicable. Study limitations and strengths are discussed. Conclusions are supported by the data. Future research directions are discussed.

Suggested edit

1. Instead of using the term ‘complementary and alternative medicine,’ consider using the term ‘traditional, complementary, and integrative medicine (TCIM).’

Reviewer #2: I am pleased to recommend the acceptance of this manuscript for publication in your esteemed journal. The study provides a comprehensive analysis of the trends and changes in the utilization of Traditional Korean Medicine (TKM) from 2013 to 2022, offering valuable insights into the structural and functional transformations within TKM.

The key findings, including the significant increase in inpatient services of TKM hospitals and medical expenses, as well as the diversification of diagnostic and treatment methods, highlight the evolving nature of TKM. These observations are crucial for understanding the current landscape of TKM and its integration into the broader healthcare system.

While the study acknowledges some limitations, its contributions to the field are substantial. The call for future research to include all TKM-related data and to examine sociocultural and clinical factors influencing TKM utilization is well-justified. Such comprehensive studies are essential for policymakers, healthcare providers, and researchers aiming to promote the effective integration of TKM and improve public health outcomes in Korea.

Overall, the manuscript is well-structured, and the conclusions are supported by the data presented. I believe that this study will contribute significantly to the ongoing discussions about the role of traditional medicine in modern healthcare systems.

Thank you for the opportunity to review this manuscript.

Reviewer #3: This is a simple and well written manuscript describing the evaluation and outcomes of evaluation of an EBP continuing education course for naturopathic doctors in Canada. It highlights a gap in that evaluation of EBP training for NDs may not have previously been performed, whereas it has for doctors of medicine. My comments are minor and based on questions arising while reading the manuscript - that if addressed may lead to future readers having fewer questions about what and why.

Introduction:

If possible, add a little more context.

What is the level of education of ND in Canada?

What is the extent of research skills learnt currently during ND training and in training over the past 20 years?

Only licensed practitioners were eligible - but what is the eligibility criteria for licensing?

How many licensed Canadian NDs are there? (this is also relevant later when interpreting the success of recruitment and reach of the course.

Methods

Please add the institutions through which the course was developed delivered. It came as a surprise in the Discussion that the course was for fee paying practitioners. I assume then that it was delivered via the Canadian College of Naturopathic Medicine. Was this the first time the course was run or just the first time it was evaluated?

Lines 157-163. Would it work better to have the second sentence first and edit the first sentence to fit. As it is, I'm reading this as: EBASE as been used to assess changes - but only for chiropractic and nursing. But it has been administered to a far broader group of disciplines. This was confusing and had me going back to re-read the paragraph from 144 explaining EBASE. Is the difference that it is mostly used as a one off point in time measure - whereas it has been used on multiple occasions with nurses and chiros to measure change over time?

From line 164, notes that EBASE has self-report bias, so a 16 question quiz was administered. It reads as though this quiz was instead of EBASE. It's only at line 191 that it is clear that both are used. Suggest adding to the Fig 1 flow chart to include of the various recruitment and data collection steps AND time points, i.e.name the assessment tools and when administered e.g. week 0, w1, follow-up etc. Also, later, make sure use of 'w' for week is explained (lines 286-289).

One line describing qualitative data analysis is insufficient and the Results do not appear to be Braun and Clarke style thematic analysis - but more like a structured Framework analysis approach.

Results

Does the geographical local match distribution of NDs in Canada? i.e. are there very few in the provinces other than the most populous Ontario and BC?

Quebec has high population. Was lack of attendance from there due to language or other reasons? This should be noted. (This perhaps for Discussion)

Table 4. This does not look like an output from Braun and Clark style thematic analysis. It appears to be a very simple, structured framework analysis based on the evaluation questions where the 'themes' might be better labeled as 'Course element' - or some such.

Why is there nothing on most and least helpful resources in the table? Not likely an accidental omission - but without anything in the box it raises the question that it could be accidental. Suggest adding something in the empty boxes, even if only "evaluation questions did not elicit this data".

Discussion

While course enrolment does seem high given the short time of advertising, without knowing the number of licenced NDs, saying it is high is subjective. As suggested earlier, provide number of licenced NDs as denominator on which to judge success of recruitment. Can you explain the low number of participants from other provinces?

The Discussion expands on and contextualizes Results, but it doesn't reflect upon what this evaluation contributes to the field, it is lacking in direction to others who may wish to establish a similar course or evaluate something similar. Is there more published elsewhere on how the course was designed and developed? It's said that the codesign was a strength, but this is fairly meaningless when What is included on that in the intro/methods is cursory. e.g. at lines 75-76 I expected references 19,20,21 to be citing publications that described how the course was created - but they do not.

As the authors have not described how much research training is on offer to NDs in training, it's not possible to judge the importance or utility of this training.

I think the Discussion could condense some of what is currently included and reflect on why or how increased EBP knowledge and skill is helpful to the naturopathic profession, why others perhaps ought to pursue this, and on relevance to naturopaths or other T&CM practitioners in Canada or elsewhere.

Conclusion

The conclusion is appropriate given the current discussion. If suggestions made to strengthen the description of the course creation and Discussion are taken up, the conclusion should be edited accordingly.

The supplementary material is really useful.

Thank you.

6. PLOS authors have the option to publish the peer review history of their article (what does this mean? ). If published, this will include your full peer review and any attached files.

**Do you want your identity to be public for this peer review?** For information about this choice, including consent withdrawal, please see our Privacy Policy .

Reviewer #1: No

Reviewer #2: No

Reviewer #3: No

---

## [Author Response · Author response to Decision Letter 1]

23 Apr 2025

Reviewer #1: PONE-D-25-02800 "Evaluation of an Evidence-Based Practice Continuing Education Course for Canadian Naturopathic Doctors".

Comments to the Authors

Thank you for your work on this manuscript. This research will add to the literature in a number of different areas. The manuscript is well-written and meets the journal’s publication criteria.

Originality

This is an original piece of work that can add to the literature in the context of designing courses that focus on evidence-based practice (EBP) in the context of continuing education for naturopathic doctors.

Introduction

Relevant background information and a rationale for the study are provided. Study aims are clearly stated.

Methods

The Methods section is clearly organised and provides relevant information under each section. Study measures are clearly described. Detailed information of the analysis process is provided.

Results

The results are clearly outlined and answer the research aims. The Tables and Figure support the results.

Discussion

The main findings of the study are discussed, and some links are made to earlier research that is applicable. Study limitations and strengths are discussed. Conclusions are supported by the data. Future research directions are discussed.

Suggested edit

1. Instead of using the term ‘complementary and alternative medicine,’ consider using the term ‘traditional, complementary, and integrative medicine (TCIM).’

• Thank you for this suggestion. CAM has been switched to TCIM on page 2 and 7

Reviewer #3: This is a simple and well written manuscript describing the evaluation and outcomes of evaluation of an EBP continuing education course for naturopathic doctors in Canada. It highlights a gap in that evaluation of EBP training for NDs may not have previously been performed, whereas it has for doctors of medicine. My comments are minor and based on questions arising while reading the manuscript - that if addressed may lead to future readers having fewer questions about what and why.

Introduction:

If possible, add a little more context.

What is the level of education of ND in Canada?

• Thank you for highlighting this omission. Added to page 3

What is the extent of research skills learnt currently during ND training and in training over the past 20 years?

• Great point. While it is hard to find precise information about this progression over time and supporting citations, we have attempt to clarify this on page 3.

Only licensed practitioners were eligible - but what is the eligibility criteria for licensing?

• Added to page 3

How many licensed Canadian NDs are there? (this is also relevant later when interpreting the success of recruitment and reach of the course.

• This is an excellent point. Added to page 3

Methods

Please add the institutions through which the course was developed delivered. It came as a surprise in the Discussion that the course was for fee paying practitioners. I assume then that it was delivered via the Canadian College of Naturopathic Medicine. Was this the first time the course was run or just the first time it was evaluated?

• Yes it was delivered by CCNM’s continuing education department, for the first time. Added to the top of page 5. It is also stated in the recruitment section.

Lines 157-163. Would it work better to have the second sentence first and edit the first sentence to fit. As it is, I'm reading this as: EBASE as been used to assess changes - but only for chiropractic and nursing. But it has been administered to a far broader group of disciplines. This was confusing and had me going back to re-read the paragraph from 144 explaining EBASE. Is the difference that it is mostly used as a one off point in time measure - whereas it has been used on multiple occasions with nurses and chiros to measure change over time?

• Yes, your understanding is correct – it has been used in many cross-sectional studies involving a range of professions but has only been used to assess change (following an educational intervention) in 2 professions. The paragraph has been revised according to your suggestion.

From line 164, notes that EBASE has self-report bias, so a 16 question quiz was administered. It reads as though this quiz was instead of EBASE. It's only at line 191 that it is clear that both are used. Suggest adding to the Fig 1 flow chart to include of the various recruitment and data collection steps AND time points, i.e.name the assessment tools and when administered e.g. week 0, w1, follow-up etc. Also, later, make sure use of 'w' for week is explained (lines 286-289).

• This has been modified to make it clear that the quiz was administered in addition to the EBASE. Fig 1 has been revised to clearly show the data collected at each step and the time point.

• W has been changed to ‘week’ for clarity.

One line describing qualitative data analysis is insufficient and the Results do not appear to be Braun and Clarke style thematic analysis - but more like a structured Framework analysis approach.

• Thank you for this suggestion. You are correct. The approach that was taken was consistent with a structured framework analysis. We have adjusted the manuscript accordingly.

Results

Does the geographical local match distribution of NDs in Canada? i.e. are there very few in the provinces other than the most populous Ontario and BC?

• Thank you for highlighting this – it has been added to the discussion section

Quebec has high population. Was lack of attendance from there due to language or other reasons? This should be noted. (This perhaps for Discussion)

• Naturopathic medicine is not regulated in Quebec. We’ve added this to the discussion.

Table 4. This does not look like an output from Braun and Clark style thematic analysis. It appears to be a very simple, structured framework analysis based on the evaluation questions where the 'themes' might be better labeled as 'Course element' - or some such.

• Thank you for this suggestion. You are correct. The approach that was taken was consistent with a structured framework analysis. We have adjusted the manuscript accordingly.

Why is there nothing on most and least helpful resources in the table? Not likely an accidental omission - but without anything in the box it raises the question that it could be accidental. Suggest adding something in the empty boxes, even if only "evaluation questions did not elicit this data".

• This has been added to the table

Discussion

While course enrolment does seem high given the short time of advertising, without knowing the number of licenced NDs, saying it is high is subjective. As suggested earlier, provide number of licenced NDs as denominator on which to judge success of recruitment. Can you explain the low number of participants from other provinces?

• A statement has been added about the geographic distribution of participants in the Feasibility section of the Discussion.

The Discussion expands on and contextualizes Results, but it doesn't reflect upon what this evaluation contributes to the field, it is lacking in direction to others who may wish to establish a similar course or evaluate something similar. Is there more published elsewhere on how the course was designed and developed? It's said that the codesign was a strength, but this is fairly meaningless when What is included on that in the intro/methods is cursory. e.g. at lines 75-76 I expected references 19,20,21 to be citing publications that described how the course was created - but they do not.

• Reference 17 is a paper that we published describing the co-design process. It provides a fair bit of detail about the process we undertook and the element of the course that were identified for inclusion.

As the authors have not described how much research training is on offer to NDs in training, it's not possible to judge the importance or utility of this training.

• Added in introduction and referred to in final section of discussion

I think the Discussion could condense some of what is currently included and reflect on why or how increased EBP knowledge and skill is helpful to the naturopathic profession, why others perhaps ought to pursue this, and on relevance to naturopaths or other T&CM practitioners in Canada or elsewhere.

• A subsection has been added to the discussion titled ‘Importance’

Conclusion

The conclusion is appropriate given the current discussion. If suggestions made to strengthen the description of the course creation and Discussion are taken up, the conclusion should be edited accordingly.

The supplementary material is really useful.

Thank you.

• Thank you for your detailed and thoughtful comments. We feel that these revisions have substantially improved the quality of the manuscript

---

## [Editor Report · Decision Letter 1]

25 Apr 2025

Evaluation of an Evidence-Based Practice Continuing Education Course for Canadian Naturopathic Doctors

PONE-D-25-02800R1

Dear Dr. Cooley,

We’re pleased to inform you that your manuscript has been judged scientifically suitable for publication and will be formally accepted for publication once it meets all outstanding technical requirements.

Kind regards,

Jenny Wilkinson, PhD

Academic Editor

PLOS ONE
---

## [Editor Report · Acceptance letter]

PONE-D-25-02800R1

PLOS ONE

Dear Dr. Cooley,

I'm pleased to inform you that your manuscript has been deemed suitable for publication in PLOS ONE. Congratulations! Your manuscript is now being handed over to our production team.

Kind regards,

on behalf of

Dr Jenny Wilkinson

Academic Editor

PLOS ONE